# GRADUAL STOCHASTIC GRADIENT DESCENT: FROM SIGNSGD TO SGD VIA $\ell_p$ NORM

**Jinghui Yuan, Jiachen Liu,**[*] **Feiping Nie**[†]

School of Artificial Intelligence, Optics and Electronics (iOPEN),
Northwestern Polytechnical University,
Xi'an 710072, P.R. China
`yuanjh@mail.nwpu.edu.cn;liujiachen@mail.nwpu.edu.cn;feipingnie@gmail.com`

## ABSTRACT

The research community has long sought an optimizer that converges as quickly as Adam in the early stage while achieving the strong generalization of SGD in the later stage. In this paper, we present a novel and feasible approach toward this goal. Recent studies have shown that Adam can be viewed as a smoothed version of sign Stochastic Gradient Descent (signSGD), i.e., the steepest descent under an $\ell_\infty$ norm ball constraint, whereas stochastic gradient descent can be regarded as the steepest descent under an $\ell_2$ norm ball. Inspired by this perspective, we propose Gradual Norm Optimization framework and design Gradual Stochastic Gradient Descent algorithm (GSGD), which enables the optimizer to smoothly transition from sign-based stochastic gradient descent in the early phase to standard stochastic gradient descent at the end. Gradual Stochastic Gradient Descent requires modifying only a single line of the original SGD implementation. We conduct preliminary evaluations of GSGD on CIFAR-10 datasets, and the experimental results show that it exhibits fast convergence in the early stage, while retaining the generalization in the later stage.

## 1 INTRODUCTION

Adam (Kingma, 2014) and SGD are currently the most mainstream optimizers in the deep learning community and are widely used in fields such as computer vision (Wang et al., 2025), natural language processing (Liu et al., 2025), and multimodal large-scale models (Achiam et al., 2023). In practice, each has its own advantages. The prevailing view is that Adam converges faster than SGD in the early stage, while SGD exhibits better generalization performance than Adam (Zhou et al., 2020). However, this relationship is not universal: recent work has shown that adaptive optimizers such as Adam and Muon can in fact generalize better than SGD in other settings, including

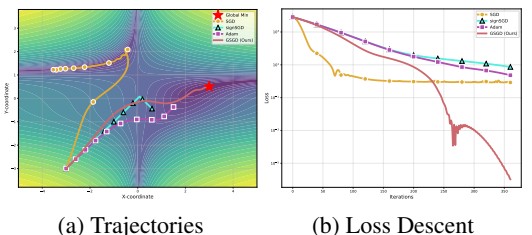

(a) Trajectories     (b) Loss Descent

Figure 1: On the Beale function, comparing different optimizers with the same parameters, GSGD shows faster speed and lower loss, significantly outperforming other optimizers. Using a logarithmic curve on the y-axis, GSGD only exhibits minor oscillations at convergence.

transformer-based architectures (Zhang et al., 2024; Vasudeva et al., 2025b;a). Consequently, the research community has long been exploring how to design an optimizer that behaves like Adam in the early phase and inherits the advantages of SGD in the later phase (Zhuang et al., 2020).

Recently, Bernstein & Newhouse has attempted to unify well-known optimizers such as Adam, SGD, and Muon (Jordan et al., 2024) within a single framework. Specifically, when the moving average is removed, Adam can be viewed as the steepest descent under an $\ell_\infty$ norm ball ($\beta = 0$),

---

[*]Co-first author.

[†]Corresponding author.

while SGD can be regarded as the steepest descent under an $\ell_2$ norm ball. This is equivalent to seeking an optimizer whose initial behavior approximates the steepest descent under an $\ell_\infty$ norm ball and gradually transitions to the steepest descent under an $\ell_2$ norm ball in the later stage.

One of the simplest, most natural ways to guarantee a smooth transition during optimization is to apply a homotopy transformation (Hatcher, 2005) on the norm. Specifically, at the $t$-th optimization step, we use an $\ell_{p(t)}$ norm ball to determine the optimization direction. As the iteration from 1 to the final step $N$, the $\ell_{p(t)}$ norm should smoothly transition from the $\ell_\infty$ norm to the $\ell_2$ norm. We refer to this approach of adapting the norm ball to the needs of the optimization process as *Gradual Norm Optimization (GNO)*. In particular, we propose *Gradual Stochastic Gradient Descent*, abbreviated as *GSGD*. We conduct a convergence analysis of GSGD and prove that when the objective function is $L$-smooth, GSGD converges to a stationary point at a rate of $\mathcal{O}\left(\frac{1}{\sqrt{T}}\right)$ after Equation (4). Figure 1 illustrates the optimization trajectories on the Beale function, visually confirming this behavior.

We demonstrated the efficacy of GSGD on the CIFAR-10 dataset (Krizhevsky et al., 2009) and the Resnet (He et al., 2016). The results reveal that GSGD mirrors the trajectory of signSGD during the initial phase and smoothly transitions to match the generalization capability of SGD in the final phase. The achieved accuracy underscores GSGD's potential as a competitive optimizer. Given its simplicity, requiring only a one line code modification over SGD, we encourage the community to jointly explore the limits of GSGD and the broader Gradual Norm Optimization framework in large-scale settings.

## 2 GRADUAL STOCHASTIC GRADIENT DESCENT

### 2.1 ALGORITHM IDEAS

In this section, we introduce the ideas of Gradual Stochastic Gradient Descent. First, the general expression of the steepest descent method is given by Equation (1).

$$\Delta w = \arg\min_{\widetilde{\Delta w}} \left[ g^T \widetilde{\Delta w} + \frac{\lambda}{2} \|\widetilde{\Delta w}\|_p^2 \right] = -\frac{\|g\|^\dagger}{\lambda} \arg\max_{\|t\|_p=1} g^T t \tag{1}$$

Here, $\Delta w$ denotes the update, $g \in \mathbb{R}^n$ is the current gradient, $\|\cdot\|_p$ denotes the $\ell_p$ norm, and $\|\cdot\|^\dagger$ denotes the dual norm. Different choices of $p$ lead to fundamentally different optimizers. In particular, when $p = 2$, $\Delta w = -\frac{1}{\lambda} g$, which corresponds to standard gradient descent. When $p = \infty$, $\Delta w = -\frac{\|g\|_1}{\lambda} \text{sign}(g)$, which corresponds to sign gradient descent, where $\text{sign}(\cdot)$ denotes the element-wise sign of a vector. For a general $\ell_p$ norm, we can obtain a similar result, as given in Equation (2). Here, $q$ satisfies $\frac{1}{p} + \frac{1}{q} = 1$, $\frac{1}{\xi}$ is a scalar, and the main term is given by $\text{sign}(g) \odot |g|^{q-1}$.

$$\Delta w = -\frac{\|g\|_q}{\lambda \|g\|_q^{q/p}} \cdot \text{sign}(g) \odot |g|^{q-1} = -\frac{1}{\xi} \cdot \text{sign}(g) \odot |g|^{q-1} \tag{2}$$

Bernstein & Newhouse points out that when the moving average coefficient $\beta$ of Adam is set to $0$, its update rule becomes Equation (3). This is equivalent to sign gradient descent, i.e., the steepest descent under the $\ell_\infty$ norm.

$$\Delta w = -\eta \frac{g}{\sqrt{g \odot g}} = -\eta \cdot \text{sign}(g) \tag{3}$$

Therefore, to find an optimizer that behaves like Adam in the early stage and like SGD in the later stage is equivalent to seeking an optimizer that uses the $\ell_\infty$ norm direction in the early phase and the $\ell_2$ norm direction in the later phase.

However, there exists a simple and highly natural way to achieve this. As the training process $t$ progresses, we set a time-varying parameter $p(t)$ and, at each optimization step, use the $\ell_{p(t)}$ norm to guide the optimization. By allowing $\ell_{p(t)}$ to smoothly transition from the $\ell_\infty$ norm at the beginning of training to the $\ell_2$ norm in the final stage, the optimizer behaves like Adam in the early phase and like SGD in the later phase. In other words, we only need to find a function $h(q(t))$ such that, as $t$ evolves, the term $|g|^{h(q(t))}$ in Equation (2) transitions from $|g|^0$ to $|g|^1$. That's the core idea.

---

**Algorithm 1** *Gradual Stochastic Gradient Descent*: As the optimization proceeds, the norm is smoothly transitioned from $\ell_\infty$ to $\ell_2$, causing the optimizer to behave like signSGD/Adam in the early stage and like SGD in the later stage.

---

**Require:** $\eta$: Learning rate
**Require:** $N$: Total number of iterations
**Require:** $w_0$: Initial parameter vector
**Require:** $f(w)$: Objective function
  1: $t \leftarrow 0$
  2: **while** $t < N$ **do**
  3:    $t \leftarrow t + 1, \quad g_t \leftarrow \tilde{\nabla} f(w_t)$          $\tilde{\nabla} f(w_t)$ is stochastic gradient.
  4:    $w_t \leftarrow w_{t-1} - \eta \cdot \mathrm{sign}(g_t) \odot |g_t|^{\frac{t-1}{N}}$       The only difference.
  5: **end while**
  6: **return** $w_N$

---

## 2.2 Algorithm details

In this section, we present the implementation details of GSGD. Our algorithm is shown in Algorithm 1. It is easy to see that the key difference between GSGD and SGD lies in only a single line: SGD always uses the $\ell_2$ norm to define the steepest descent, whereas GSGD employs a time-varying norm ball. In the design process, there are still two points that require attention.

First, at the $t$-th iteration, the $\ell_{p(t)}$ norm we adopt satisfies $p(t) = \frac{N}{t-1} + 1$. This design ensures that the dual norm associated with the $\ell_p$ norm changes linearly. Directly setting $p(t) = t$ is not feasible, as it would cause the exponent of $|g|$ in $\mathrm{sign}(g) \odot |g|^{q-1}$ to remain close to zero for most of the training process. As a result, the optimizer would behave like signSGD for most of the time and then rapidly decay to SGD in the later stage, which is not the desired outcome.

The second point is that when training neural networks, in order to maintain stability of the gradient magnitude and facilitate learning rate adjustment, we scale the gradient. Specifically, when updating $w_t$, we multiply $\Delta w$ by a scalar. Our recommended update formula is Equation (4).

$$w_t \leftarrow w_{t-1} - \eta \cdot \frac{\|g_t\|_2}{\|\mathrm{sign}(g_t) \odot |g_t|^{\frac{t-1}{N}}\|_2} \cdot \mathrm{sign}(g_t) \odot |g_t|^{\frac{t-1}{N}} \tag{4}$$

In addition, we can incorporate momentum and Nesterov acceleration (Nesterov, 1983; Yuan et al., 2025) in exactly the same way as in standard SGD. Specifically, after computing the gradient, one can smooth it to obtain momentum, or apply a lookahead step to achieve Nesterov acceleration. Similarly, norm regularization can be applied to the momentum or to the lookahead gradient.

## 2.3 Algorithm Analysis

GSGD has the same time and space complexity as SGD; the only overhead is an element-wise exponent computation, which is $\mathcal{O}(n)$ and practically negligible.

Furthermore, considering the convergence analysis, we present Theorem 1, which shows that GSGD possesses the same convergence properties and convergence rate as SGD.

**Theorem 1.** *Suppose the objective function $f(w)$ is L-smooth, the gradient magnitude is normalized according to Equation (4) and step size $\eta = \frac{1}{\sqrt{T}}$. Suppose the stochastic gradient $g_t$ is an unbiased estimator of the true gradient $\nabla f(w_t)$, and its variance is bounded by $\sigma^2$. We assume that $\Delta w$ is a descent direction as in Equation (8), which holds under Gradual Gradient Descent. Then:*

$$\min_{t \in \{1,...,T\}} \mathbb{E}[\|\nabla f(w_t)\|_2^2] \sim \mathcal{O}\left(\frac{1}{\sqrt{T}}\right) \tag{5}$$

This theorem provides several insights. If changing the norm used for the metric does not affect the convergence rate, then it may not be necessary to use the steepest descent under a fixed norm throughout all stages of optimization. This observation further motivates the study of gradual norm optimization, suggesting that other more advanced optimization algorithms can be extended to the gradual norm setting, potentially leading to improved performance.

Table 1: Comparison of average performance metrics (%) at the final training epoch

| Method | Train Acc (%) | Val Acc (%) | Test Acc (%) | Train Loss |
|--------|---------------|-------------|--------------|------------|
| Adam | 98.54±0.01 | 91.09±0.19 | 90.65±0.23 | 0.0503±0.0005 |
| SGD | 97.93±0.22 | 91.52±0.22 | 91.18±0.20 | 0.0713±0.0047 |
| signSGD | 97.58±0.16 | 90.52±0.01 | 90.06±0.18 | 0.0663±0.0041 |
| GSGD | **98.93**±0.03 | **91.85**±0.07 | **91.19**±0.07 | **0.0415**±0.0014 |

## 3 EXPERIMENTS

This preliminary work focuses on establishing the theoretical framework and providing a proof of concept on CIFAR-10. Our comparison methods include signSGD, Adam, and SGD. We use ResNet-20 as the network architecture, with configurations consistent with those in (Bernstein et al., 2018).

For hyperparameter selection, GSGD is tuned over the following ranges: learning rate $\eta \in \{0.001, 0.01\}$, momentum hyperparameter $\beta \in \{0.0, 0.5, 0.9\}$, and weight decay $\lambda \in \{0.001, 0.01, 0.1\}$. To more thoroughly tune the baseline methods, we search their hyperparameters over a broader range: learning rate $\eta \in \{0.001, 0.01, 0.1, 1\}$, momentum hyperparameter $\beta \in \{0.0, 0.5, 0.9\}$, and weight decay $\lambda \in \{0.001, 0.01, 0.1\}$. The remaining hyperparameters follow (He et al., 2016), which are considered to be favorable for the SGD algorithm.

The dataset split is also kept consistent with the original signSGD paper: the 60k CIFAR-10 samples are divided into $\{45k/5k/10k\}$ for the training, validation, and test sets, respectively. Figure 3 shows the validation accuracy under large-scale hyperparameter searches, where the best-performing configuration on the validation set for each algorithm is selected for the final test run. We visualize the evolution of training accuracy, validation accuracy, test accuracy and the loss curves for each algorithm under the best hyperparameter settings selected on the validation set in Figure 2.

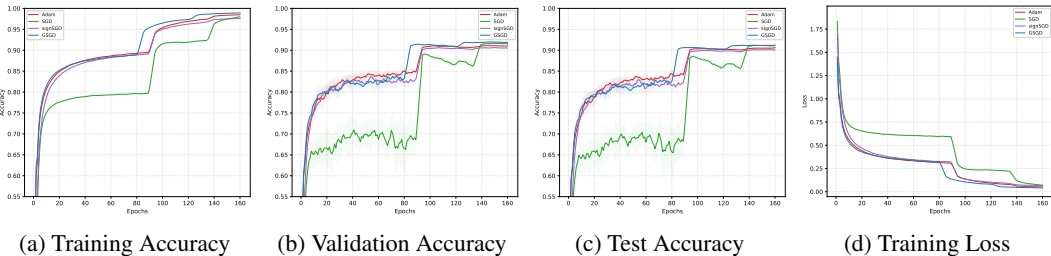

| (a) Training Accuracy | (b) Validation Accuracy | (c) Test Accuracy | (d) Training Loss |

Figure 2: Averaged performance curves over three runs under the best hyperparameter setting.

We report the averaged final test results in Table 1. Combined with the performance curves, we observe that GSGD consistently achieves the highest training accuracy and the lowest loss, while its test and validation accuracies are comparable to those of SGD and higher than those of signSGD and Adam. Moreover, during the early stage of training, GSGD exhibits performance improvements comparable to Adam and superior to SGD. These results demonstrate that GSGD successfully achieves the intended behavior: behaving similarly to signSGD/Adam in the early training phase, while approaching SGD in the later stage.

## 4 CONCLUSION

Our work proposes the Gradual Norm Optimization (GNO) framework and correspondingly introduces the Gradual Stochastic Gradient Descent (GSGD) algorithm. By dynamically adjusting the $\ell_p$ norm constraint during training, GSGD successfully combines the exploration capability of signSGD/Adam with the generalization ability of SGD. We establish its convergence properties and validate its practical potential through experimental results.

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

# A  APPENDIX

## A.1  PROOF OF THEOREM 1

The objective function $f(w)$ is $L$-smooth, meaning its gradient is Lipschitz continuous. For any $x, y \in \mathbb{R}^d$, we have Equation (6).

$$f(y) \leq f(x) + \langle \nabla f(x), y - x \rangle + \frac{L}{2} \|y - x\|_2^2 \tag{6}$$

The stochastic gradient $g_t$ is an unbiased estimator of the true gradient $\nabla f(w_t)$, and its variance is bounded by $\sigma^2$, which means Equation (7).

$$\mathbb{E}[g_t] = \nabla f(w_t), \quad \mathbb{E}[\|g_t - \nabla f(w_t)\|_2^2] \leq \sigma^2 \tag{7}$$

From this assumption, we have the property regarding the second moment that $\mathbb{E}[\|g_t\|_2^2] \leq \|\nabla f(w_t)\|_2^2 + \sigma^2$.

The GSGD update rule in Equation (4) generates a valid descent direction. Specifically, since $p(t) \geq 2$, the steepest descent direction based on the $\ell_{p(t)}$ norm forms an acute angle with the negative gradient. Combined with the normalization in Equation (4), we assume there exists a constant $c > 0$ such that the expected update satisfies:

$$\mathbb{E}_t[\langle \nabla f(w_t), w_{t+1} - w_t \rangle] \leq -c\eta \|\nabla f(w_t)\|_2^2 \tag{8}$$

Let the update rule be $w_{t+1} = w_t + \Delta w_t$. According to $L$-smooth, expanding at $w_t$:

$$f(w_{t+1}) \leq f(w_t) + \langle \nabla f(w_t), \Delta w_t \rangle + \frac{L}{2} \|\Delta w_t\|_2^2 \tag{9}$$

The update step is explicitly normalized. The magnitude of the update is scaled such that $\|\Delta w_t\|_2 = \eta \|g_t\|_2$. Substituting this into the quadratic term of the inequality:

$$f(w_{t+1}) \leq f(w_t) + \langle \nabla f(w_t), \Delta w_t \rangle + \frac{L\eta^2}{2} \|g_t\|_2^2 \tag{10}$$

Taking the expectation $\mathbb{E}_t[\cdot]$ conditioned on $w_t$ on both sides, and we have Equation (11):

$$\mathbb{E}_t[f(w_{t+1})] \leq f(w_t) - c\eta \|\nabla f(w_t)\|_2^2 + \frac{L\eta^2}{2}(\|\nabla f(w_t)\|_2^2 + \sigma^2) \tag{11}$$

Rearranging the terms to isolate $\|\nabla f(w_t)\|_2^2$:

$$\mathbb{E}_t[f(w_{t+1})] \leq f(w_t) - \left(c\eta - \frac{L\eta^2}{2}\right) \|\nabla f(w_t)\|_2^2 + \frac{L\eta^2 \sigma^2}{2} \tag{12}$$

Summation and Convergence Rate. Assume the step size $\eta$ is small enough such that $\eta \leq \frac{c}{L}$, which implies $c\eta - \frac{L\eta^2}{2} \geq \frac{c\eta}{2}$. Moving the gradient term to the left side:

$$\frac{c\eta}{2} \|\nabla f(w_t)\|_2^2 \leq f(w_t) - \mathbb{E}_t[f(w_{t+1})] + \frac{L\eta^2 \sigma^2}{2} \tag{13}$$

Taking the total expectation and summing from $t = 1$ to $t = T$:

$$\frac{c\eta}{2} \sum_{t=1}^{T} \mathbb{E}[\|\nabla f(w_t)\|_2^2] \leq f(w_1) - \mathbb{E}[f(w_{T+1})] + \frac{TL\eta^2 \sigma^2}{2} \tag{14}$$

Let $f^*$ be the global minimum. Then $f(w_1) - \mathbb{E}[f(w_{T+1})] \leq f(w_1) - f^* = \Delta F$. Dividing both sides by $T$:

$$\frac{1}{T} \sum_{t=1}^{T} \mathbb{E}[\|\nabla f(w_t)\|_2^2] \leq \frac{2\Delta F}{c\eta T} + \frac{L\eta \sigma^2}{c} \tag{15}$$

Substituting the step size $\eta = \frac{1}{\sqrt{T}}$ as stated in Theorem 1:

$$\min_t \mathbb{E}[\|\nabla f(w_t)\|_2^2] \leq \frac{1}{T} \sum_{t=1}^{T} \mathbb{E}[\|\nabla f(w_t)\|_2^2] \leq \frac{2\Delta F}{c\sqrt{T}} + \frac{L\sigma^2}{c\sqrt{T}} \tag{16}$$

Thus, we conclude that the convergence rate is following:

$$\min_{t \in \{1,\dots,T\}} \mathbb{E}[\|\nabla f(w_t)\|_2^2] \sim \mathcal{O}\left(\frac{1}{\sqrt{T}}\right) \tag{17}$$

This concludes the proof.

## A.2 ADDITIONAL RESULTS

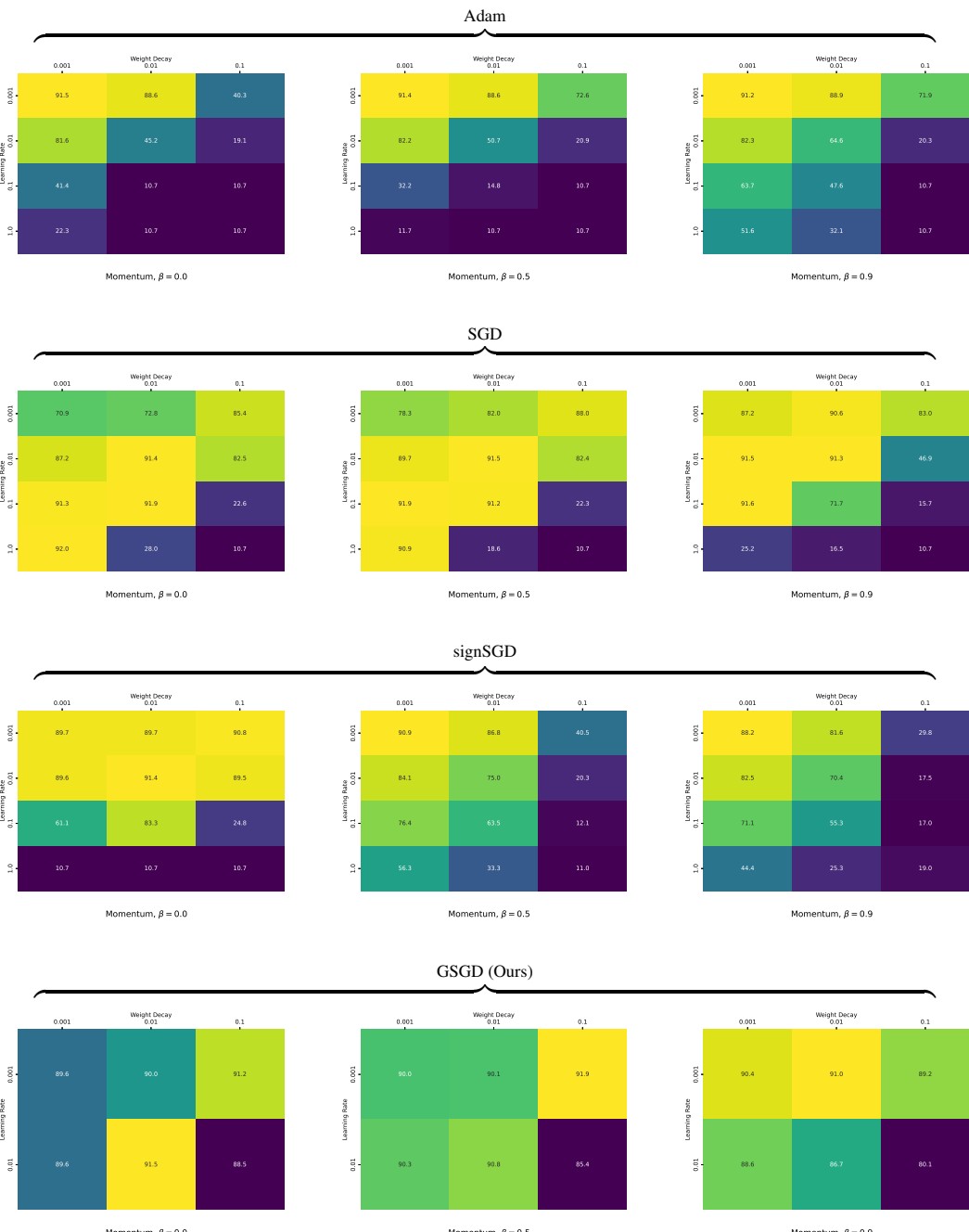

Figure 3: Highest validation accuracy for different optimizers across varying hyperparameters.

The heatmap results show that GSGD significantly outperforms signSGD and Adam, and under hyperparameter settings favorable to SGD, it achieves performance close to that of SGD.

## A.3 LIMITATIONS

Several limitations of this work should be acknowledged. First, the experimental evaluation is restricted to ResNet-20 on CIFAR-10; validation on transformer architectures and language modeling tasks is an important next step, which we are actively pursuing. Second, the paper lacks compari-

---

**Algorithm 2** *Gradual Adam*: A seamless transition from Adam to SGD. The denominator's exponent $q$ decays from 0.5 to 0.

---

**Require:** $\eta$: Learning rate
**Require:** $N$: Total number of iterations
**Require:** $\beta_1, \beta_2$: Moment coefficients (default: 0.9, 0.999)
**Require:** $w_0$: Initial parameters, $f(w)$: Objective function
 1: $t \leftarrow 0$, $m_0 \leftarrow 0$, $v_0 \leftarrow 0$
 2: **while** $t < N$ **do**
 3: $\quad t \leftarrow t + 1$, $\quad g_t \leftarrow \nabla f(w_{t-1})$
 4: $\quad m_t \leftarrow \beta_1 m_{t-1} + (1 - \beta_1) g_t$
 5: $\quad v_t \leftarrow \beta_2 v_{t-1} + (1 - \beta_2) g_t \odot g_t$
 6: $\quad \hat{m}_t \leftarrow m_t/(1 - \beta_1^t)$, $\quad \hat{v}_t \leftarrow v_t/(1 - \beta_2^t)$
 7: $\quad q \leftarrow 0.5 \times \max(0, 1 - \frac{t-1}{N})$ $\qquad$ Dynamic Power: $0.5 \rightarrow 0$
 8: $\quad w_t \leftarrow w_{t-1} - \eta \cdot \frac{\hat{m}_t}{\hat{v}_t^q + \epsilon}$
 9: **end while**
10: **return** $w_N$

---

son with normalized gradient descent (NGD) and normalized momentum descent (NMD) baselines; disentangling the benefit of norm scheduling from that of gradient normalization alone is an open empirical question we plan to address. Third, the convergence proof relies on the assumption that the GSGD update is a descent direction Equation (8), and providing a tighter, self-contained theoretical analysis—ideally with provable generalization guarantees—remains future work. Finally, whether dedicated learning rate decay strategies further improve GSGD is an important direction for future research.

## A.4 GRADUAL ADAM

Analogous to Gradual Stochastic Gradient Descent, one can similarly construct a Gradual Adam algorithm, which can be viewed as a smoothed version of GSGD. Specifically, we consider taking an element-wise $q$ th power of the second-moment estimate $v$ maintained by Adam (Chen et al., 2018). When $q = \frac{1}{2}$, this recovers standard Adam, which, as discussed earlier, corresponds to the steepest descent under the $\ell_\infty$ norm. When $q = 0$, it reduces to standard SGD. Therefore, by allowing $q$ to vary with the optimization steps and interpolate over the interval $(\frac{1}{2}, 0)$, we obtain the core idea of GAdam.

## A.5 FEATURE WORKS

In the future, we will attempt to adapt Gradual Stochastic Gradient Descent properties to the spherical cautious optimizers (Yuan et al., 2026) and apply it to multiple optimization domains (Yuan & Nie, 2026). At the same time, we will try to decouple $|g|^{h(q(t))}$ from the time parameter $t$, and explore the use of random exponents in the hope of obtaining certain regularization properties.

