# OpenReview forum: "Gradual Stochastic Gradient Descent: from signSGD to SGD via $\ell_p$ Norm"
_ICLR.cc/2026/Workshop/Sci4DL — Sci4DL 2026_

### Official Review · Reviewer_yr8g · 2026-02-17

**Fit:** 2
**Significance:** 2
**Confidence:** 2

**Summary:**

The authors propose a novel optimizer, GSGD, which smoothly transitions from signSGD to SGD during training. This is achieved by scheduling the exponent of the absolute gradient from 0 to 1 within the parameter update step before multiplying it by the gradient’s sign. The authors motivate their algorithm by combining the advantages of Adam (fast convergence in early epochs) and SGD (superior generalization in later epochs).

**Strengths:**

The idea is great and simple. The paper is well structured, easy to follow and well motivated.

**Suggestions:**

- I would not cite LeCun et al. (2002) when discussing simple SGD, as SGD was not proposed in that paper. It is standard textbook knowledge; you do not need a reference.
- What do the markers represent in Fig. 1? Why are there none for GSGD? If these are the steps, why are they so different for each optimizer? Generally, much of this example looks constructed. Is the learning rate cherry-picked for GSGD?
- In Eq. 1, the $\Delta w$ inside the arg min should have a tilde or similar.
- The discussion regarding "Directly setting $p(t)=t$" is incorrect. This would cause the optimizer to evolve from SGD to signSGD (when starting at $t=2$) and not the other way around.
- The scaling factor in Eq. 4 appears out of nowhere. I do not understand how it is motivated. The factor causes an additional learning rate scheduling, which should be disentangled from the proposed gradient exponent scheduling. Additionally, the sign has no effect in the l2-norm. CSGD should be compared against learning rate scheduling.
- The language requires significant improvement.
- It appears you did not cover the optimal learning rates for Adam. In Fig. 3 the best performance is found at the lowest tested learning rate, so you should test lower ones.
- Even for a workshop paper, the experimental evaluation with only a ResNet on CIFAR10 is somewhat weak.

---

### Official Review · Reviewer_NaDo · 2026-02-28

**Fit:** 2
**Significance:** 2
**Confidence:** 3

**Summary:**

The paper proposes Gradual Stochastic Gradient Descent (GSGD) algorithm, that interpolates between signSGD-like updates (more similar to Adam) in the initial stages of training to SGD-like updates during later stages of training. This is done by employing steepest descent updates with respect to $\ell_p$-norm and varying $p$ from $\infty$ to $2$ as training progresses. Preliminary results on CIFAR-10 dataset show that the algorithm converges faster in initial stages and performs similar or slightly better than SGD.

**Strengths:**

The proposed algorithm is very simple and can be implemented easily.

It is supported by a convergence guarantee, although the proof is quite simple.

The presented experiments (although limited) support the main claims.

**Suggestions:**

**Main Weaknesses:**

1. One issue is that the premise that “SGD generalizes better than Adam” is not at all well-motivated. It has been shown in several recent works (e.g., [1-3]), that adaptive optimizers like Adam and Muon in fact generalize better than SGD in several settings. If the sole focus of the paper is in-distribution generalization with CNNs trained on image classification datasets, then it should specify and cite other relevant papers to support this motivation.

2. Secondly, the experimental results are extremely limited. The paper only includes results with ResNet-20 model trained on CIFAR-10. The authors should include more results with other model and task families including experiments with transformers trained on language datasets.

3. Additionally, the algorithm uses normalized steepest descent updates. So, the right baselines in this case would be normalized gradient descent (NGD, or respectively, its momentum based version, normalized momentum descent, i.e., NMD). The algorithm should be renamed appropriately and in several places in the paper, the scaling should be taken care of. In the current experiments, it’s unclear whether using NGD or NMD (layer-wise normalization) would not give similar speed-ups in the initial stage like Adam or signGD; comparison with GD for fixed training steps is not fair.

4. Finally, the theoretical results in the paper can be strengthened significantly. For instance, can the authors show that using this gradual norm optimization framework provably improves generalization in some setting?

**Minor suggestions:**

1. Figure resolution should be improved, currently font size and line thickness are small and things are hard to see.

2. Some typos: Cifar in place of CIFAR in the abstract, Lines 61-62 should be rephrased, Table 1 caption should be rephrased.

**References:**

1. Why Transformers Need Adam: A Hessian Perspective, Zhang et al., NeurIPS 2024.

2. The rich and the simple: On the implicit bias of Adam and SGD, Vasudeva et al., NeurIPS 2025.

3. How Muon’s spectral design benefits generalization, Vasudeva et al., ICLR 2026.

---

### Meta-Review · Area_Chair_1vmB · 2026-02-28

**Recommendation:** Accept
**Confidence:** 4

**Metareview:**

Both reviewers gave positive feedback, so I'll accept it.

My main note here is about the scientific value here. You can roughly split fundamental DL research into on-the-whiteboard theory (classical learning theory, optimization theory, etc.), applied work (e.g. trying out ideas while training nets + measuring performance), and between these, empirical science: testing ideas, measuring things other than performance, and trying to understand what's going on. This paper hits the first two, but the latter one's really where we need to go with the study of optimizers: empirically, we know that things like Adam and Muon work, and sure, we can prove convergence theorems, but we don't really know that they're telling us anything meaningful. It'd be nice to see some empirical effort here to understand why and how GSGD changes optimization dynamics.

---

### Decision · Program_Chairs · 2026-03-02

Accept